# Radiation-Induced Structural Changes of *Miscanthus* Biomass

**Xiao-Jun Su** [1,2]**, Chun-Yan Zhang** [1]**, Wen-Jia Li** [1]**, Feng Wang** [1]**, Ke-Qin Wang** [3]**, Yun Liu** [4,5,6,7,*] **and Qing-Ming Li** [1,2,*]

[1] College of Food Science and Technology, Hunan Agricultural University, Changsha 410128, China; suxiaojun5606@163.com (X.-J.S.); zhangchy95@163.com (C.-Y.Z.); liwenjia9301@163.com (W.-J.L.); wanglaofeng@163.com (F.W.)
[2] Hunan Engineering Laboratory for Alcohol Fuels from Biomass, Changsha 410128, China
[3] Institute of Nuclear Agricultural Science and Space Breeding, Hunan Academy of Agriculture Sciences, Changsha 410125, China; wkq6412@163.com
[4] College of Life Science and Technology, Beijing University of Chemical Technology, Beijing 100029, China
[5] School of Nuclear Technology and Chemistry & Biology, Hubei University of Science and Technology, Xianning 437100, China
[6] Hubei Key Laboratory of Radiation Chemistry and Functional Materials, Hubei University of Science and Technology, Xianning 437100, China
[7] Hubei Engineering Research Center for Fragrant Plants, Hubei University of Science and Technology, Xianning 437100, China
[*] Correspondence: liuyun@mail.buct.edu.cn (Y.L.); liqmemail@163.com (Q.-M.L.)

**Abstract:** Efficient pretreatment is a prerequisite for lignocellulosic biomass biorefinery due to the structure of lignocellulose. This study is a first-time investigation into the structural changes of *Miscanthus* biomass treated with $^{60}$Co γ-ray irradiation in different doses up to 1200 kGy. The structural properties of the treated sample have been systematically characterized by FTIR, thermogravimetric analysis (TGA), XRD, gel permeation chromatography (GPC), a laser particle size analyzer, SEM, an atomic force microscope (AFM), and NMR. The results show that irradiation treatment can partially destroy the intra- or inter-molecular hydrogen bonds of biomass. Irradiation treatment can also reduce particle size, narrow the distribution range, as well as increase the specific surface area of biomasses. Noticeably, the TGA stability of the treated biomass decreases with increasing absorbed doses. To respond to these structural changes, the treated biomass can be easily hydrolyzed by cellulases with a high yield of reducing sugars (557.58 mg/g biomass), much higher than that of the untreated sample. We conclude that irradiation treatment can damage biomass structure, a promising strategy for biomass biorefinery in the future.

**Keywords:** *Miscanthus* biomass; irradiation pretreatment; recalcitrant; structural properties; enzymatic hydrolysis

## 1. Introduction

As a C$_4$ perennial grass, *Miscanthus* is an important potential energy crop in China and Europe. It is characterized by higher adaptability, bio-production, and fiber content, as well as lower ash content and input requirements in comparison to conventional crops [1,2]. The chemical composition of *Miscanthus* includes cellulose (370–501 g/kg dry solids), hemicellulose (283–354 g/kg dry solids), and lignin (68.7–127.6 g/kg dry solids) [3]. Biofuels and biochemicals from *Miscanthus* have been comprehensively studied in the past few years. However, the recalcitrant structure of biomass has become one of the main factors restricting biofuel production [1,4]. Cellulose and hemicellulose

constitute the whole biomass and are firmly linked with lignin molecules through covalent and hydrogenic bonds, which make the biomass structure extremely strong and difficult to pretreat [5,6]. To enhance cellulose hydrolysis, the method of biomass pretreatment should efficiently damage the recalcitrant biomass structure [7–9]. In recent years, many researchers have reported a number of excellent pretreatment strategies, including milling, alkali, acid, hydrothermal and ammonia explosion methods, as well as biological degradation [1,3,9–12].

Recently, more efforts have focused on irradiation pretreatment of biomass using mild temperatures, no water washing, and minimal use of undesirable inhibitory products [13]. Moreover, irradiation pretreatment can significantly damage the recalcitrant structure of lignocellulosic biomass, especially degrading cellulose structure, which facilitates the downstream enzymatic saccharification [14]. In our previous studies, the effect of irradiation pretreatment on the structure of microcrystal cellulose as well as hemicellulose was intensively investigated [15,16], and we demonstrated that irradiation pretreatment followed by enzymatic hydrolysis can result in high sugar yields [17]. Therefore, irradiation pretreatment may be one of the most promising methods to overcome this recalcitrance.

To date, little information has been reported on the degradation mechanism of real biomass treated by irradiation. In this work, irradiation pretreatment was employed to degrade the structure of *Miscanthus* to improve its downstream enzymatic hydrolysis. The effect of different absorbed doses up to 1200 kGy on recalcitrant structures is illustrated to better understand the degradation mechanism of irradiation treatment.

## 2. Materials and Methods

### 2.1. Materials

*Miscanthus floridulus* was gathered from a local experimental base in Hunan Agricultural University, China. After it was dried in an oven at 60 °C for 8 h, the samples were ground and sieved through 40 mesh(the particle size of sample is approx.425 μm). The feedstock was stored at room temperature for the following experiments.

### 2.2. $^{60}$Co γ-Ray Irradiation Pretreatment of Miscanthus

The procedures of irradiation treatment and enzymatic hydrolysis were conducted according to the method reported by Liu et al. [14]. Briefly, the irradiation treatment experiments were carried out on a $^{60}$Co γ-ray irradiation device at $1.85 \times 10^{16}$ Bq in the Hunan Irradiation Center (Liuyang City, China). Approximately 200 g of grounded dry *Miscanthus* biomass was put in a glass bottle, and then irradiated at a $^{60}$Co-γ radiation source intensity of $9.99 \times 10^{15}$ Bq at 2.0 kGy h$^{-1}$ dose rate. The specific levels of absorbed doses were fixed at 0 (untreated as blank control), 400, 600, 800, 1000, and 1200 kGy.

### 2.3. Chemical Compositions Analyses

The chemical compositions of *Miscanthus*, including the content of moisture, cellulose, lignin, and reducing sugar before and after irradiation treatment were determined using the analytical methods provided in the GB/T 2677 standard. The content of hemicellulose was calculated by subtracting cellulose from holocellulose [18]. Each experimental result was repeated in triplicate.

### 2.4. Structural Properties of the Irradiated Miscanthus

### 2.4.1. Gel Permeation Chromatography (GPC)

The GPC method was employed to determine the molecular weight (MW) distribution of irradiated *Miscanthus*, and the detailed procedure of GPC was described in a previous study [16].

### 2.4.2. X-Ray Diffraction Analysis (XRD)

The crystalline phase of irradiated *Miscanthus* cellulose was demonstrated by a D/max 2500 diffractometer (Rigaku Corporation, Tokyo, Japan). The conditions were: acceleration voltage 40 kW and current 30 mA. The Cu-K$\alpha$ wavelength was 1.54 Å, grade range $2\theta$ ranged from 10° to 40° with the step size of 0.002°. The crystallinity index (*CrI*) of cellulose was calculated using Equation (1) [18]

$$CrI\ (\%) = \frac{I_{002} - I_{am}}{I_{002}} \times 100 \qquad (1)$$

where *CrI* is the crystallinity index, $I_{002}$ is for the highest peak intensity at $2\theta = 22.6°$, and $I_{am}$ is the intensity diffraction for amorphous cellulose at $2\theta = 16.0°$.

### 2.4.3. Solid State $^1$H and $^{13}$C nuclear magnetic resonance (NMR)

$^1$H and $^{13}$C NMR measurements were conducted on a Bruker AVANCE III-HD 600 spectrometer (Bruker, Priyanka City, Germany). Their resonance frequencies were 600.1 and 150.9 MHz, respectively. $^1$H and $^{13}$C NMR spectra were both recorded using a 4 mm magic angle spinning (MAS) probe equipped with a spinning rate of 14 kHz. Other conditions of the $^1$H NMR spectra were a $\pi/2$ pulse length of 2.57 µs and a recycle delay of 5 s. Other conditions of $^{13}$C NMR spectra were a contact time of 2 ms and a recycle delay of 5 s.

### 2.4.4. Fourier Transform Infrared spectroscopy (FTIR)

FTIR measurements of *Miscanthus* samples were carried out on a Prestige-21 FTIR instrument (Shimadzu, Japan). All spectra were recorded in the range of 4000 to 400 cm$^{-1}$ with an accumulation of 128 scans at 4 cm$^{-1}$ resolution. Each sample was in conjunction with potassium bromide (KBr) powder and pressed inside a hydraulic press for spectroscopic analytical sample preparation.

### 2.4.5. Particle size distribution and specific surface Area (SSA)

SSA measurement was performed on a laser particle size analyzer (Mastersizer 2000, Malvern, Worcestershire, U.K.). The sample was uniformly dispersed in sodium hexametaphosphate solution, and then loaded into the analyzer for measurement.

### 2.4.6. Scanning electron microscope (SEM)

The morphology of *Miscanthus* was observed by a JSM-6360LV SEM (Japan Electron Optics Laboratory Co., Ltd., Tokyo, Japan). The samples were first sputter-coated with gold prior to observation by SEM [19].

### 2.4.7. Atomic force microscope (AFM)

The dimension morphology of *Miscanthus* biomass after irradiation treatment was imaged using AFM (Bruker, Priyanka City, Germany) through ScanAsyst mode with a scanning frequency of 1.49 Hz. All images of height, amplitude and phase were simultaneously obtained in tapping mode with a MPP-11100 etched silicon probe, of which the nominal frequency and nominal spring constant were set at 300 kHz and 40 N /m, respectively [20].

### 2.4.8. Thermogravimetric snalysis (TGA)

TGA thermal stability of the *Miscanthus* biomass was determined on a TGA Q50 analyzer (Waters Co., Milford, MA, USA). The N$_2$ with a flow rate of 50 mL/min was used as the carrier gas. The temperature ranged from room temperature to 900 °C with a heating rate of 30 °C/min.

### 2.4.9. Degree of Polymerization (*DP*)

The *DP* of the *Miscanthus* biomass was measured through the viscosity variance using Ubbelohde viscometry at 25 ± 0.5 °C. An extrapolation method was used to calculate the intrinsic viscosity ($\eta_t$) of the sample. Therefore, the values of $\eta_t$ and *DP* were calculated according to Equations (2) and (3):

$$\eta_t = \frac{t - t_0}{t_0} \tag{2}$$

$$DP = \frac{2000 \times \eta_T}{W \times (1 + 0.29 \times \eta_T)} \tag{3}$$

where $t_0$ and $t_i$ are the initial and end times (min) for the cupriethylenediamine solution to run through the capillary, respectively, and *W* is the mass weight (g) of the sample.

### 2.5. Enzymatic Hydrolysis of the Irradiated Miscanthus

The irradiated *Miscanthus* biomass was used for enzymatic hydrolysis according to the procedure described by Su et al. [17]. After hydrolysis, the reducing sugars were determined by HPLC. The conditions of HPLC were a temperature of 65 °C, and the mobile phase was 5 mM $H_2SO_4$ at a flow rate of 0.6 mL/min [14].

### 2.6. Statistical Analysis

The statistical analyses were determined using SPSS 22.0 (IBM, Armonk, NY, USA), and the final data are expressed by average ± SD.

## 3. Results and Discussion

### 3.1. Effect of Irradiation Treatment on Chemical Compositions of Miscanthus *Biomass*

The composition changes of *Miscanthus* biomass after different absorbed doses are presented in Table 1. As shown in Table 1, the main components of *Miscanthus* are cellulose, hemicellulose, and lignin, and their total content accounts for 89.7%. In comparison with the compositions of giant reed and Chinese silvergrass reported in the literature, *Miscanthus* has the same contents of holocellulose and lignin as giant reed, and the total content is higher than Chinese silvergrass [18]. After irradiation treatment, the contents of cellulose and hemicellulose of *Miscanthus* decreased with the increase in absorbed doses up to 1200 kGy, and irradiation treatment had little influence on lignin content. That is, the content of lignin before and after irradiation treatment was almost stable. These phenomena were confirmed by other researches [18,21]. The structure of destroyed cellulose can improve enzymatic hydrolysis during downstream saccharification [13]. To illustrate why irradiation has a significant effect on cellulose and hemicellulose but little effect on lignin, the biomass structure was comprehensively elucidated by FTIR, TGA, XRD, GPC, a laser particle size analyzer, SEM, AFM, and NMR in this work.

**Table 1.** Effect of different absorbed doses on the components of *Miscanthus*.

| Absorbed Dose (kGy) | 0 (Untreated) | 400 | 600 | 800 | 1000 | 1200 |
|---|---|---|---|---|---|---|
| Cellulose (%) | 36.60 | 37.20 | 33.90 | 33.10 | 31.00 | 23.00 |
| Hemicellulose (%) | 31.80 | 17.80 | 11.60 | 10.50 | 7.40 | 4.90 |
| Lignin (%) | 21.30 | 23.10 | 21.70 | 22.00 | 23.00 | 19.70 |
| Total (%) | 89.70 | 78.10 | 67.20 | 65.60 | 61.40 | 47.60 |

### 3.2. Effect of Irradiation Treatment on Enzymatic Hydrolysis of Miscanthus

Table 2 shows the changes in particle size, distribution, and special surface area of *Miscanthus* biomass after irradiation treatment. As shown in Table 2, the Sauter mean diameter D [3, 2] and the volume average particle diameter D [4, 3] greatly decreased with the increase in absorbed doses.

For the untreated sample, the values of D [3, 2] and D [4, 3] were 23.808 and 221.005 µm, respectively, higher than those of irradiated samples. For instance, after irradiation treatment at 1200 kGy, the values of D [3, 2] and D [4, 3] were 7.357 and 20.099 µm, respectively. In addition, the values of d (0.1), d (0.5), and d (0.9) for untreated samples were 18.026, 153.465, and 547.317 µm, respectively, whereas for the irradiated samples at 1200 kGy, these values significantly reduced to 2.772, 18.423, and 39.082 µm, respectively. Therefore, an absorbed dose-dependent particle size for irradiated biomass was observed in this work. In detail, with the increase in absorbed doses, particle size distribution moved toward small particles. This showed that the irradiation treatment can remarkably affect both the particle sizes and their distribution due to the destruction of the stubborn structure caused by irradiation. As shown in Table 2, the SSA of the untreated *Miscanthus* was 0.252 $m^2 g^{-1}$, whereas for irradiated samples, SSA showed an absorbed dose-dependent increase and the maximum value was 0.815 $m^2 g^{-1}$ at 1200 kGy. These results are in good agreement with our previous works [19,21]. We demonstrated that the increase in SSA enhances the accessibility of enzymes to cellulose substrate, resulting in the improvement of cellulose digestibility.

**Table 2.** Effect of absorbed doses on particle size and SAA of *Miscanthus* biomass.

| Absorbed Dose (kGy) | SSA ($m^2$/g) | D [3, 2] (µm) | D [4, 3] (µm) | d (0.1) (µm) | d (0.5) (µm) | d (0.9) (µm) |
|---|---|---|---|---|---|---|
| 0 (Untreated) | 0.252 | 23.808 | 221.005 | 18.026 | 153.465 | 547.317 |
| 400 | 0.256 | 23.432 | 203.056 | 14.963 | 131.620 | 496.540 |
| 600 | 0.315 | 19.053 | 169.749 | 4.397 | 107.316 | 462.257 |
| 800 | 0.319 | 18.806 | 169.307 | 4.641 | 105.537 | 461.163 |
| 1000 | 0.642 | 9.343 | 33.060 | 2.924 | 21.181 | 90.421 |
| 1200 | 0.815 | 7.357 | 20.099 | 2.772 | 18.423 | 39.082 |

Note: SSA, specific surface area; D [3, 2], Sauter mean diameter; D [4, 3], the volume mean particle diameter; d (0.5), median diameter; d (0.1), 10% diameter; d (0.9), 90% diameter.

To further confirm the above-mentioned hypothesis, the irradiated biomass underwent enzymatic hydrolysis to reduce sugar production, and the results are shown in Figure 1. In comparison with the untreated *Miscanthus* sample, the irradiated sample at over 600 kGy had a higher level of methylene blue adsorption (596.99 µg/g) and a higher reduced sugar yield (557.58 mg/g). The methylene blue adsorption, a factor of enzyme accessibility to the substrate, slowly increased from 555.18 to 596.99 µg/g in the tested absorbed doses. It increased sugar reduction from 118.27 to 557.58 mg/g. A reasonable explanation is that the accessibility of enzymes increased after the biomass was treated by irradiation, indicating irradiation pretreatment is an effective method to improve the efficiency of enzymatic hydrolysis of a biomass. This phenomenon is in good agreement with that reported by Beardmore et al. [22], who demonstrated that SSA improvement of sulfite pulp increases the efficiency of enzymatic hydrolysis.

*3.3. Influence of Irradiation Treatment on Structural Properties of* Miscanthus *Biomass*

3.3.1. *DP*, *CrI*, and Molecular Weight Distribution

The effect of irradiation treatment on molecular weight distribution, *DP*, and *CrI* of *Miscanthus* biomass was investigated, and the results are shown in Table 3. When the biomass was irradiated from 0 to 1200 kGy, the *DP* values decreased from 366,225 to 11,354, indicating that irradiation treatment can destroy the biomass structure. Moreover, the values for Mw(the weight average molecular mass) and Mn (the number average molecular mass) decreased with the increase in absorbed doses. Briefly, the Mw and Mn values of the untreated samples (0 kGy) were 542,342 and 45,544 Da, respectively, whereas for the irradiated samples at 1200 kGy, the values of Mw and Mn decreased to 150,821 and 30,237 Da, respectively. In comparison with the untreated sample, the value of Mw/Mn decreased after irradiation treatment. However, the value of Mw/Mn showed no absorbed dose dependence due to

the similar decrease variance in Mw and Mn values. We found that the *Miscanthus* macromolecular structure was degraded after irradiation treatment [23]. These structural changes in biomass are helpful for enzymatic hydrolysis [24]. Table 3 shows that the *CrI* values of irradiated samples showed a dose-dependent decreasing tendency, which also improved the enzymatic hydrolysis efficiency. The change in *CrI* values demonstrated that irradiation treatment may damage the crystalline structure of cellulose, which was confirmed by TGA analysis in the following experiment.

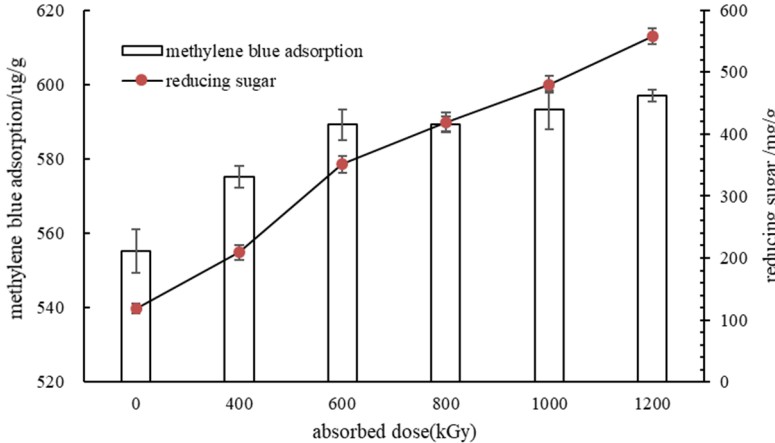

**Figure 1.** Effect of absorbed dose on reducing sugars yield and methylene blue adsorption.

**Table 3.** Influence of absorbed doses on molecular weight distribution, *DP*, and *CrI* of *Miscanthus* biomass.

| Dose/kGy | 0 | 400 | 600 | 800 | 1000 | 1200 |
|---|---|---|---|---|---|---|
| Mw (Da) | 542,342 | 228,791 | 168,982 | 160,859 | 151,213 | 150,821 |
| Mn (Da) | 45,544 | 37,981 | 36,273 | 34,793 | 33,565 | 30,237 |
| Mw/Mn | 14.952 | 5.024 | 4.449 | 4.623 | 4.505 | 4.988 |
| *DP* | 366,225 | 44,230 | 30,947 | 20,569 | 18,200 | 11,354 |
| *CrI* (%) | 37.86 | 33.26 | 28.92 | 29.03 | 30.02 | 28.43 |

Note: Mw, the weight average molecular mass; Mn, the number average molecular mass; Mw/Mn, the polydispersity index; *DP*, the degree of polymerization; *CrI*, the crystallinity index.

### 3.3.2. TGA Measurement

The TGA thermal stability of biomass usually has two obvious narrow peaks: one is the degradation peak of hemicellulose, the other is the peak of cellulose. However, the TGA peak of lignin is very broad and partly overlaps hemicellulose and cellulose [25]. The TGA profiles of the irradiated *Miscanthus* biomass are depicted in Figure 2. There are two peaks ($T_{max L}$ and $T_{max H}$) observed from TGA curves at doses of 400, 600, and 800 kGy, but only a single peak ($T_{max L}$) at 1000 and 1200 kGy. The disappearance of the high-temperature peak ($T_{max H}$) was caused by the side chains cleavage of biomass under the higher absorbed doses. In addition, a small shift in $T_{max L}$ value to a lower temperature (Table 3) occurred when absorbed dose increased up to 1200 kGy, indicating the backbone of the biomass was degraded by the irradiation treatment. The phenomenon does not align with the *CrI* values of cellulose in Table 3.

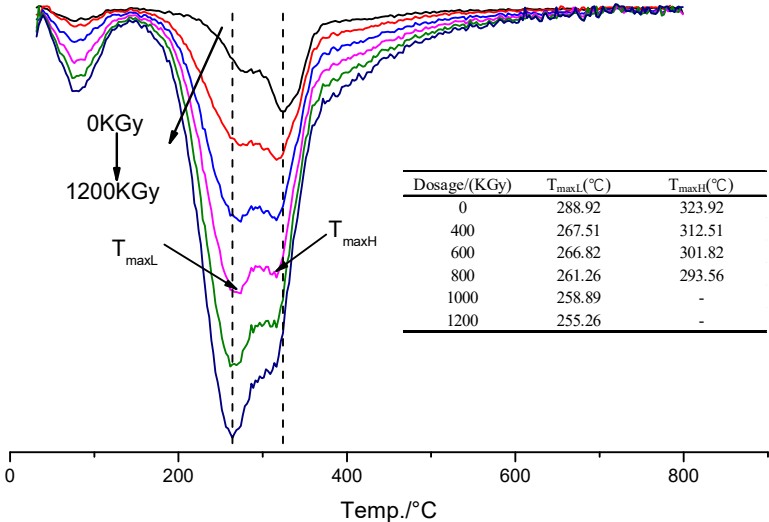

| Dosage/(KGy) | $T_{maxL}$(°C) | $T_{maxH}$(°C) |
|---|---|---|
| 0 | 288.92 | 323.92 |
| 400 | 267.51 | 312.51 |
| 600 | 266.82 | 301.82 |
| 800 | 261.26 | 293.56 |
| 1000 | 258.89 | - |
| 1200 | 255.26 | - |

**Figure 2.** Effect of absorbed doses on the thermal stability profiles of *Miscanthus* biomass.

### 3.3.3. XRD Analysis

XRD measurement was conducted to assess the crystalline phase of *Miscanthus* biomass under different absorbed doses. As observed in Figure 3, the *CrI* peaks at ~16° and 22° lattices remained stable without new lattice generation after irradiation treatment. As described in Table 3, the *CrI* values showed an absorbed dose-dependent decrease, indicating the crystalline area of cellulose was destroyed to some extent. The *CrI* values of irradiated *Miscanthus* declined slightly from 37.86% to 28.43% under the absorbed doses from 0 to 1200 kGy (Table 3). The effect of irradiation treatment on cellulose crystalline phase is dependent on absorbed dosage and the biomass species [16,26]. Huang et al. [26] demonstrated that the crystalline phase cleavage by irradiation treatment is contributed by the enzymatic hydrolysis of biomass.

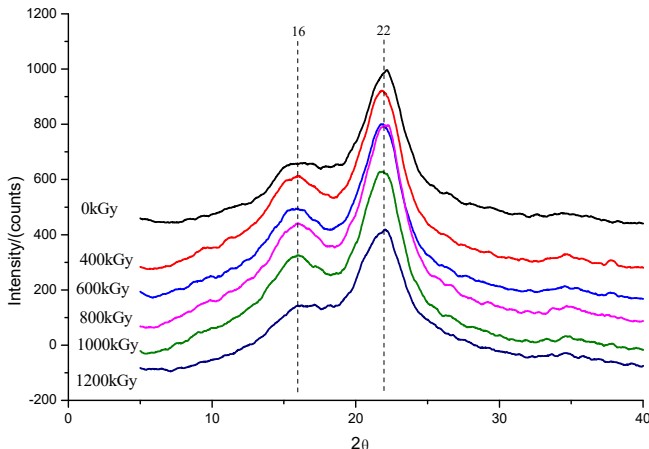

**Figure 3.** XRD patterns of the *Miscanthus* biomass irradiated with different doses.

### 3.3.4. FTIR Analysis

The FTIR of *Miscanthus* biomass after irradiation treatment was investigated, and the spectra are shown in Figure 4. Three characteristic peaks of holocellulose are: 1730 cm$^{-1}$ is ascribed to C=O carbonyl group, 1372 cm$^{-1}$ is ascribed to –C–C–, and 1237 cm$^{-1}$ is ascribed to C–O– of acetyl group, as observed in Figure 4. With the increase in absorbed doses, the intensity of the three peaks increased, indicating that the structural damage degree of *Miscanthus* fibers increases with the increase in the absorbed dose. The changes of the typical peaks of the guaiacyl and syringyl lignin vibration at 1230–1515 cm$^{-1}$ also showed absorbed dose-dependence. Simultaneously, the peak at 910 cm$^{-1}$

ascribed to the C–O–C group in the epoxide guaiacyl lignin was so small that the change in its height was not significant, which is good agreement with the data of wheat straw treated by irradiation and diluted acid [27].

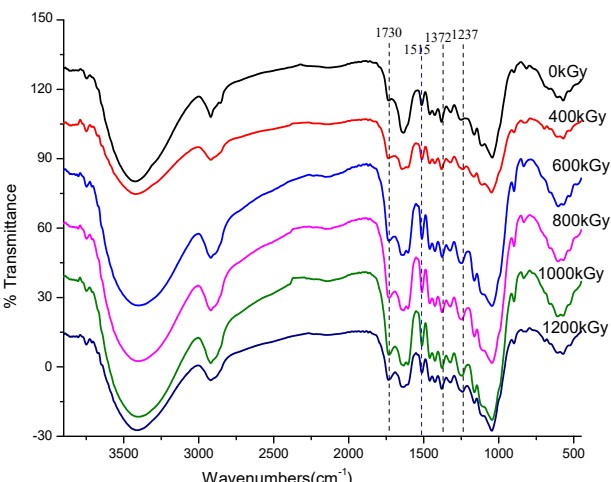

**Figure 4.** FTIR spectra of *Miscanthus* irradiated with different absorbed doses.

### 3.3.5. [1]H and [13]C NMR Analysis

[1]H and [13]C NMR analyses were carried out to evaluate the chemical structural changes of *Miscanthus* biomass after irradiation treatment, and the data are shown in Figure 5. For the untreated sample (0 kGy), the chemical shift at 5 ppm of [1]H NMR profiles was ascribed to the $H^+$ proton signal peak of hemicellulose, whereas in the cases of irradiated samples, the [1]H proton signal peak was apparently enlarged and shifted to 5.2 ppm with the increase in absorbed doses from 400 to 1200 kGy. From the [13]C NMR profiles, the peaks intensity of the glucosyl unit at 21.3, 74.6, and 88.7 ppm reduced with the increase in absorbed doses. However, the intensity of the peak at 56.6 ppm increased with the increase in the absorbed dose. These small variations in carbon chemical shifts of irradiated samples may lead to intermolecular C–C bond cleavage in backbone structure. In combination with the FTIR, and [1]H and [13]C-NMR profiles, we concluded that inter-molecular hydrogen bonds and carbon–carbon bond cleavage of biomass are caused by high absorbed doses [13,19].

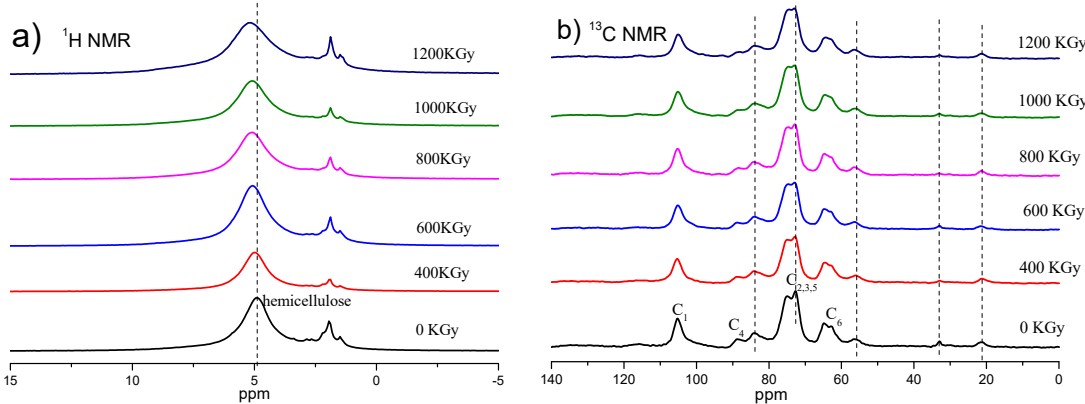

**Figure 5.** [1]H NMR (**a**) and [13]C NMR (**b**) profiles of *Miscanthus* irradiated with different doses.

### 3.3.6. SEM Analysis

The morphology of *Miscanthus* biomass after irradiation treatment was investigated by SEM, and the images are presented in Figure 6. The morphology of untreated *Miscanthus* (0 kGy) showed a smoothly compacted and ordered surface. After irradiation treatment, the surface morphology of

samples displayed many small fragments, and the structure was rough and irregular. The degree of structural damage grew stronger with the increase in absorbed doses up to 1200 kGy. The phenomenon shows that irradiation treatment can effectively destroy the tight structure of the *Miscanthus* biomass, which was confirmed by many researchers in literature [18,28,29]. The morphology change of cellulose by irradiation treatment will improve the accessibility of cellulase to biomass, leading to high enzymatic digestibility [14,30].

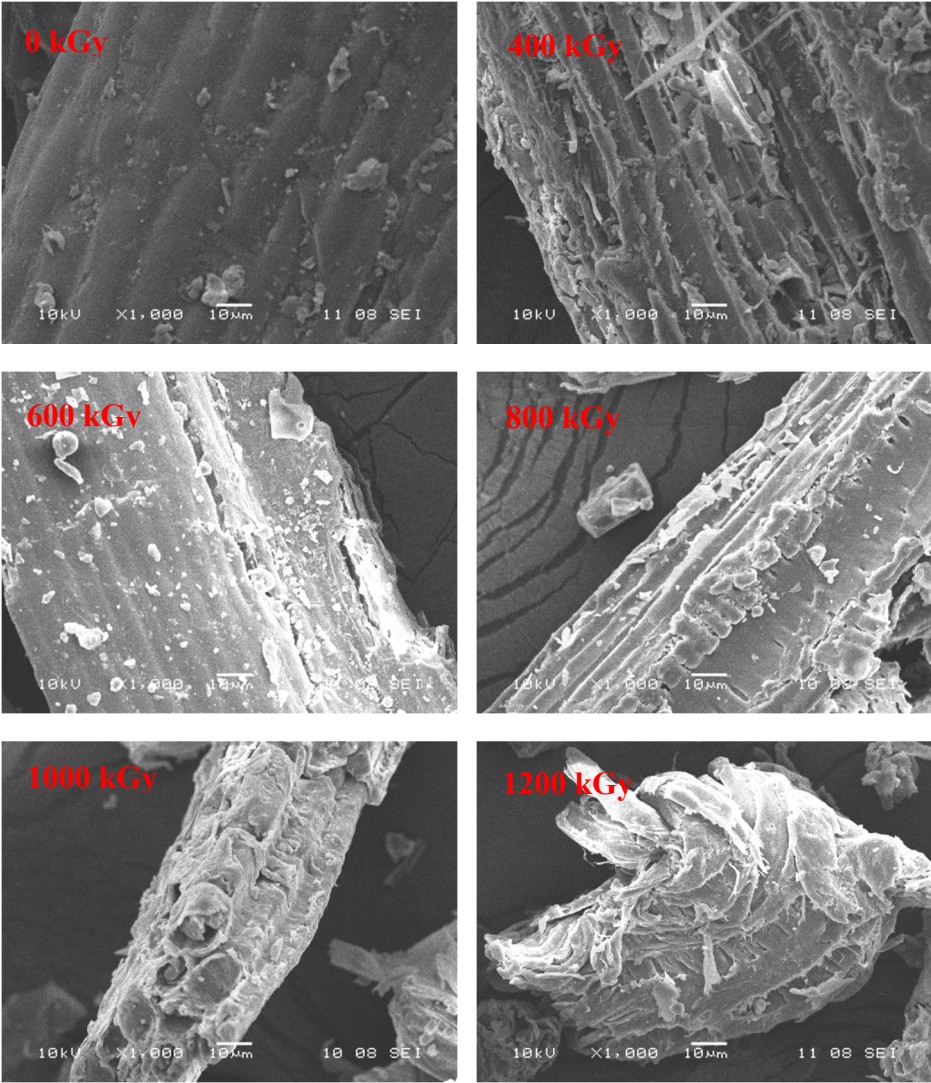

**Figure 6.** SEM photographs of untreated and irradiation-pretreated *Miscanthus* (1000×).

### 3.3.7. AFM Analysis

AFM images of untreated and irradiation treated *Miscanthus* biomass are shown in Figure 7. In the case of the untreated sample (0 kGy), the AFM image showed an intact and uniform surface, and the affinity of the AFM probe to the hydrophilic region is 46.53 nm (Figure 7a). For irradiation treated samples, their AFM images appeared non-uniform and there was a spherical surface in the amplitude phase, probably attributed to the exposure of cellulose (Figure 7b–f). The affinity of the AFM probe to the hydrophilic regions increased up to 243.67 nm. The light color of hydrophilic regions means a remarkable change in the phase image. A similar phenomenon was confirmed by other researchers in the literature [20,31]. Chundawat et al. [31] reported that the AFM probe adhered more keenly to hydrophilic areas, resulting in a greater change in the phase of the sample.

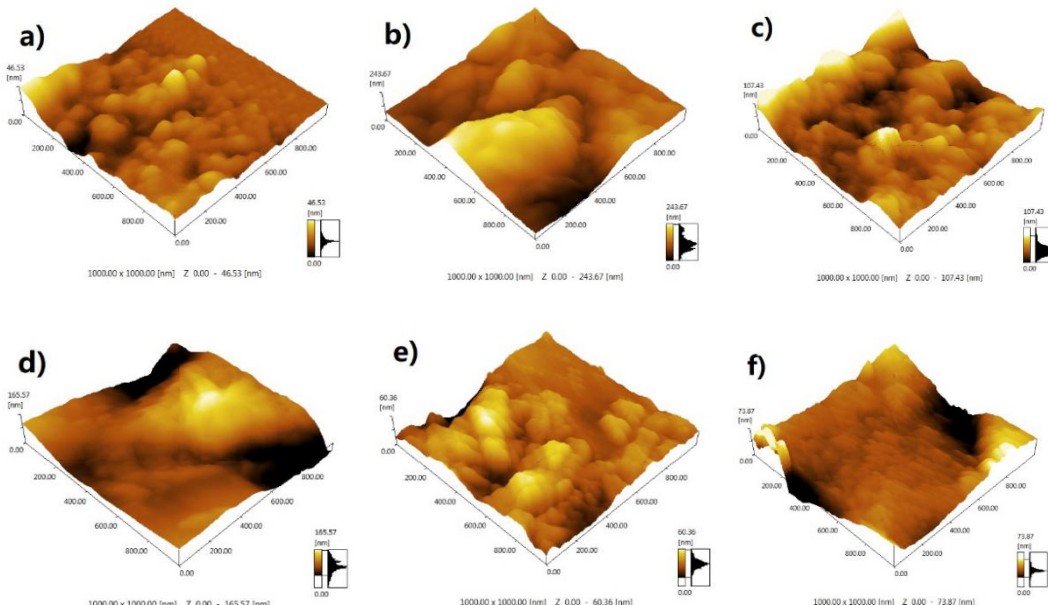

**Figure 7.** AFM images of untreated and irradiation treated *Miscanthus* biomass.

## 4. Conclusions

In summary, these findings demonstrate that γ-irradiation treatment can decrease crystalline cellulose, particle size, and *DP* while increasing the SSA of a biomass, resulting in the improvement in the cellulase accessibility to cellulose. Furthermore, irradiation treatment can substantially reduce structural and thermal stability by damaging the recalcitrant biomass structure. These structural changes in irradiated biomass will contribute to the further reduction of sugar release. The findings in this work provide an insight into the degradation mechanism of biomass by irradiation treatment, which is helpful for biomass biorefinery in the future. However, the practical application of irradiation treatment should be still discussed for its economic feasibility, which is under consideration in our laboratory.

**Author Contributions:** Conceptualization, X.-J.S. and Q.-M.L.; Methodology, C.-Y.Z., W.-J.L. and F.W.; Formal analysis, Y.L. and K.-Q.W.; Writing and checking draft manuscript, Q.-M.L., X.-J.S. and Y.L.; Visualization, Q.-M.L.; Funding acquisition, X.-J.S. All authors have read and agreed to the published version of the manuscript.

**Funding:** This work was supported by a special project for the construction of innovative provinces in Hunan Province of China (2019NK2031-2, 2019NK2031-3). This work was also funded by Hubei University of Science and Technology (2019-20KZ05).

**Conflicts of Interest:** The authors declare no conflict of interest.

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
