# Peer review of "Radiation-Induced Structural Changes of Miscanthus Biomass"

_applsci, doi:10.3390/app10031130_

Round 1

Reviewer 1 Report

I reviewed this resubmitted article a month ago.

Authors have provided reasonable response to this reviewer's comments.

Author Response

Thank you for your good comments. All authors have carefully checked the whole manuscript, and improved the English spelling to avoid the spelling errors as well as the bad sentence style. We are sure that the revised version is more readable and clearer than the origin version.

Reviewer 2 Report

The structural changes of Miscanthus biomass caused by gamma radiation was investigated using several analytical techniques.

The Introduction is much better now than in the previous version of the manuscript.

There are problems with the English of the manuscript, especially with terms. Irradiation dose is not a correct term. Correct it to absorbed dose or dose. The English of the text was corrected where I indicated the correction, however, the other parts are unchanged.

I have still conceptual problem with the manuscript.

The basic problem is that 1200 kGy is not an economical dose. This method has no chance for practical application. The samples were irradiated with 2 kGy/h dose rate meaning that 600 hours are needed for 1200 kGy absorbed dose. The dose rate can be somewhat higher but it cannot be high enough to reduce the irradiation time for a value acceptable for practical application.

It is not a new finding that the structure pf cellulose will change and its molecular mass will decrease after irradiation with 1200 kGy. The change in the crystalline structure during irradiation (not in the presence of irradiation) is also well known. Therefore, there is no new information in the manuscript.

2.4.4. Fourier Transforms Infrared Analysis (FT-IR)

Describe sample preparation method and indicate the accessory applied (ATR?) Describe the method applicable to draw quantitative conclusions from the FTIR spectra of biomass samples. In the present form I cannot accept the conclusions based on FTIR data. E.g., the peak at 910 cm-1 is so small that the change in its height is not significant.

Corrections suggested

Line 22: rephrase: in the presence of irradiation atmosphere

Line 28: intra- or inter-molecular structure of biomass – do you mean intra- or inter-molecular H-bridge bonds?

Line 32: rephrase: treated biomass can be easily enzymatic hydrolysis

Line 65: of different absorbed doses up to 1200 kGy

Line 78: at 2.0 kGy h-1.dose rate

Line 116: mode with a scanning frequency

Line 140: 3.1. Effect of absorbed dose

Line 141: after different absorbed doses

Line 142: As shown in Table 1, the main components of Miscanthus are cellulose

Line 144: rephrase: Chinese silvergrass, Miscanthus has the same biomass components content as giant reed, and higher han Chinese silvergrass[18].

Line 182: is treated by irradiation,

Line 199: rephrase: the ratio of Mw/Mn is decreasing in the presence of irradiation doses.

Line 212: rephrase: TGA decomposition of biomass

Line 216: curves at doses of 400, 600, and 800 kGy

Line225: rephrase the sentence: XRD measurement was investigated … irradiated doses

Line 227: rephrase: However,, the Crl values 228 gradually slow down with the increase of irradiation dose,

Author Response

Point 1: The structural changes of Miscanthus biomass caused by gamma radiation was investigated using several analytical techniques.

The Introduction is much better now than in the previous version of the manuscript.

There are problems with the English of the manuscript, especially with terms. Irradiation dose is not a correct term. Correct it to absorbed dose or dose. The English of the text was corrected where I indicated the correction, however, the other parts are unchanged.

Response 1: Thank you for your good comments. The term of “irradiation dose” has been corrected to “absorbed dose” in the whole manuscript.

Point 2: The basic problem is that 1200 kGy is not an economical dose. This method has no chance for practical application. The samples were irradiated with 2 kGy/h dose rate meaning that 600 hours are needed for 1200 kGy absorbed dose. The dose rate can be somewhat higher but it cannot be high enough to reduce the irradiation time for a value acceptable for practical application.

Response 2: As the reviewer description, absorbed dose of 1200 kGy is not practical application for irradiation pretreatment due to its long period time (approx. 25 days). However, irradiation pretreatment for biomass biorefinery shows many advantages, such as mild temperature, no water washing and minimal undesirable inhibitory products. Moreover, irradiation pretreatment can significantly facilitate the downstream enzymatic saccharification to yield high content reducing sugars (Bioresour Technol 2015, 182, 289-295). Therefore, irradiation pretreatment may be one of the promising methods to overcome the recalcitrance of biomass. With the better understanding of effect of irradiation pretreatment on biomass structure, we can employ integrated pretreatment proposals including acid and irradiation pretreatment with low dose to treat biomass, and make it more practical application.

Point 3: It is not a new finding that the structure of cellulose will change and its molecular mass will decrease after irradiation with 1200 kGy. The change in the crystalline structure during irradiation (not in the presence of irradiation) is also well known. Therefore, there is no new information in the manuscript.

Response 3: As we description in the mansucript, there are several articles on the effect of irradiation pretreatment on the structure of cellulose and its molecular mass. In the reported related papers, microcrystalline cellulose was typically used as experimental feedstock. To date, no information about on the effect of irradiation pretreatment on the structure of Miscanthus biomass and its molecular mass. From this point of view, this manuscript provides new insights into the change of real biomass cellulose structure and molecular mass in the presence of irradiation.

Point 4: 2.4.4. Fourier Transforms Infrared Analysis (FT-IR)

Describe sample preparation method and indicate the accessory applied (ATR?) Describe the method applicable to draw quantitative conclusions from the FTIR spectra of biomass samples. In the present form I cannot accept the conclusions based on FTIR data. E.g., the peak at 910 cm-1 is so small that the change in its height is not significant.

Response 4: For FT-IR analysis, Each sample was in conjunction with potassium bromide (KBr) powder and pressed inside a hydraulic press for spectroscopic analytical sample preparation. We are carefully checking the FT-IR spectra of treated sample, the peak at 910 cm-1 is so small that the change in its height is not significant. All these revisions are corrected in the revised version.

Corrections suggested

Line 22: rephrase: in the presence of irradiation atmosphere

Response:  This sentence has been corrected to “ treated by 60Co γ-ray irradiation”

Line 28: intra- or inter-molecular structure of biomass – do you mean intra- or inter-molecular H-bridge bonds?

Response:  You are quite right. It means intra- or inter-molecular H-bridge bonds

Line 32: rephrase: treated biomass can be easily enzymatic hydrolysis

Response: This sentence has been corrected to “the treated biomass can be easily hydrolyzed by cellulases”

Line 65: of different absorbed doses up to 1200 kGy

Response: it has been corrected in the revised version.

Line 78: at 2.0 kGy h-1.dose rate

Response: it has been corrected in the revised version.

Line 116: mode with a scanning frequency

Response: it has been corrected in the revised version.

Line 140: 3.1. Effect of absorbed dose

Response: it has been corrected in the revised version.

Line 141: after different absorbed doses

Response: it has been corrected in the revised version.

Line 142: As shown in Table 1, the main components of Miscanthus are cellulose

Response: it has been corrected in the revised version.

Line 144: rephrase: Chinese silvergrass, Miscanthus has the same biomass components content as giant reed, and higher han Chinese silvergrass[18].

Response: it has been corrected in the revised version.

Line 182: is treated by irradiation,

Response: it has been corrected in the revised version.

Line 199: rephrase: the ratio of Mw/Mn is decreasing in the presence of irradiation doses.

 Response: it has been corrected in the revised version.

Line 212: rephrase: TGA decomposition of biomass

Response: it has been corrected in the revised version.

Line 216: curves at doses of 400, 600, and 800 kGy

Response: it has been corrected in the revised version.

Line225: rephrase the sentence: XRD measurement was investigated … irradiated doses

Response: it has been corrected in the revised version.

Line 227: rephrase: However,, the Crl values 228 gradually slow down with the increase ofirradiation dose

Response: it has been corrected in the revised version.

Round 2

Reviewer 2 Report

The English of the manuscript improved a lot but check the newly added parts, there are some language problems there.

I have still conceptual problem with the manuscript.

The basic problem is that 1200 kGy is not a practically applicable dose. Fig. 1: correct axis from irradiation dose to dose.

Correct

Capture to Fig. 3: Figure 3. XRD patterns of Miscanthus biomass irradiated with different doses.

Capture to Figure 4. FT-IR spectra of Miscanthus irradiated with different absorbed doses.

Capture to Figure 5. 1H NMR (a) and 13C NMR (b) profiles of Miscanthus irradiated with different doses.

Author Response

Point 1: I have still conceptual problem with the manuscript.

The basic problem is that 1200 kGy is not a practically applicable dose. Fig. 1: correct axis from irradiation dose to dose.

Response 1: Thank you for your good comments. As the reviewer suggestion, absorbed dose of 1200 kGy is not practical application for irradiation pretreatment due to its long period time (approx. 25 days). In our present work, we are manily focusing on the effect of absorbed dose on the structure change of biomass. So, in case of conceptual experiments, we design three ranges absorbed doses (low doses 0~400kGy; medium dose 400~800kGy; high dose ≧800~1200 kGy). It is a scientific research, not a real practically applicable use. In our previous study, we demonstrated aht irradiation pretreatment for biomass biorefinery showed many advantages, such as mild temperature, no water washing and minimal undesirable inhibitory products. Moreover, irradiation pretreatment can significantly facilitate the downstream enzymatic saccharification to yield high content reducing sugars (Bioresour Technol 2015, 182, 289-295). Therefore, irradiation pretreatment may be one of the promising methods to overcome the recalcitrance of biomass. With the better understanding of effect of irradiation pretreatment on biomass structure, we can employ integrated pretreatment proposals including acid and irradiation pretreatment with low dose to treat biomass, and make it more practical application.

In addition, Figure 1 has been corrected in the revised version.

Point 2: Correct

Capture to Fig. 3: Figure 3. XRD patterns of Miscanthus biomass irradiated with different doses.

Capture to Figure 4. FT-IR spectra of Miscanthus irradiated with different absorbed doses.

Capture to Figure 5. 1H NMR (a) and 13C NMR (b) profiles of Miscanthus irradiated with different doses.

Response 2: Captures of Figure 3~5 have been corrected in the revised version.

This manuscript is a resubmission of an earlier submission. The following is a list of the peer review reports and author responses from that submission.

Round 1

Reviewer 1 Report

The study presents the research results on ‘Radiation-induced structural changes of Miscanthus biomass’. In my view, the originality of this manuscript should be emphasized more and this manuscript cannot be acceptable in its present form.

A concise abstract is required. Also, more specific descriptions of authors' finding should be added in the abstract rather than overall result of study. The abstract should state briefly the purpose of the research, the principal results and major conclusions.

There are many research results about the pretreatment by radiation. Please provide the advantage of these processes and compare to other recent research papers. Please provide the novelty of this study and compare to other recent research papers in the view of yield (conversion) and productivity.

In Table 1, Cellulose (%) and Hemicellulose (%) decreased with increasing irradiation dose. However, please explain in detail why Lignin (%) was not trending.

In Table 3, the value of Mw (Da), Mn (Da) and DP decreased with increasing irradiation dose. However, please explain in detail why Mw/Mn and CrI(%) were not trending.

The explanation of results and discussion are not enough. Overall, this paper lacks the actual discussion of experimental data. They simply described their data. A more in-depth discussion other than description is needed.

Authors should discuss the potential industrial application of this technology in an economical and technical viewpoint.

Reviewer 2 Report

The main problem is with the scientific concept of the manuscript. As it is cited in the Introduction, cellulose is a radiation degradable natural polymer. However, the samples in this work contain lignin and lignin is resistant to high-energy radiation. It acts as stabilizer absorbing the energy of irradiation. Therefore, high doses are necessary to reach any effect of the radiation on the samples. There are publications on this field which are not cited in the Introduction. Irradiation treatment has no practical application in the paper industry. Some decades earlier it was published that the dose necessary to decrease the lignin content of the wood is too high. Therefore, the application of radiation in paper industry is not economical.

The dose which is economical for practical application depends on the application itself. It can vary from 2 kGy to about 100 kGy, but 1200 kGy cannot be economical.

Although, the manuscript contains a lot of nice results measured with various techniques, due to this conceptual problem I suggest to reject this manuscript.

The English of the manuscript needs to be improved throughout the manuscript. Some corrections are shown below. The proper term in radiation chemistry is dose, or absorbed dose. Dosage, irradiation dosage, dosage rate are not correct.

Line 21: source with different doses up

Line 29: with increasing the absorbed dose.

Line 96: FT-IR measurements of Miscanthus samples were carried out